# Engagement in Muscle-Strengthening Activities Lowers Sarcopenia Risk in Older Adults Already Adhering to the Aerobic Physical Activity Guidelines

**DOI:** 10.3390/ijerph18030989

**Published:** 2021-01-22

**Authors:** Jort Veen, Diego Montiel-Rojas, Andreas Nilsson, Fawzi Kadi

**Affiliations:** School of Health Sciences, Örebro University, 702 81 Örebro, Sweden; jort.veen@oru.se (J.V.); diego.montiel@oru.se (D.M.-R.); fawzi.kadi@oru.se (F.K.)

**Keywords:** resistance exercise, muscle mass, muscle strength, ageing, protein intake, lifestyle behaviours, obesity

## Abstract

Sarcopenia in older adults is associated with a higher risk of falls, disability, loss of independence, and mortality. Current physical activity (PA) guidelines recommend engagement in muscle-strengthening activities (MSA) in addition to aerobic moderate-to-vigorous physical activity (MVPA). However, little is known about the impact of MSA in addition to adherence to the MVPA recommendation in the guidelines. The aim of the present cross-sectional study was to determine whether or not engagement in MSA is linked to sarcopenia risk in older adults who meet the PA guidelines of 150 min of MVPA per week. A total of 193 community-dwelling older men and women (65–70 years) were included in the study. A continuous sex-specific clustered sarcopenia risk score (SRS) was created based on muscle mass assessed using bioelectrical impedance analysis, handgrip strength, and five times sit-to-stand (5STS) time, assessed using standardized procedures. Adherence to PA guidelines was assessed using the Actigraph GT3x accelerometer and the EPAQ2 questionnaire. Guideline adherence to MSA twice a week was related to a significantly (*p* < 0.05) lower SRS compared to those who did not. This finding was evident after adjustment for adherence to the protein intake guideline and abdominal obesity. Similar impacts were observed for muscle mass and 5-STS but not for handgrip strength. In conclusion, guideline adherence to MSA is related to lower sarcopenia risk in older adults who already accumulate 150 weekly minutes of MVPA, which reinforces the promotion of the MSA guideline, alongside the MVPA guideline, to fight against sarcopenia progression in ageing populations.

## 1. Introduction

Sarcopenia is a progressive skeletal muscle disorder clinically defined by low levels of muscle strength and muscle mass, with physical performance as an indicator of severity [1]. Older adults with sarcopenia are at a higher risk of falls, disability, loss of independence, and mortality [2].

Given the global ageing trend, with the estimated number of people worldwide aged over 65 years reaching nearly 1.5 billion in 2050 [3], it is expected that approximately 500 million older adults will be diagnosed with sarcopenia [4]. Hence, there is an urgent need to combat the growing societal burden related to sarcopenia progression and related co-morbidities. In this context, promotion of health-enhancing physical activity is widely endorsed by major public health organizations, stipulating at least 150 weekly minutes of moderate-to-vigorous physical activity (MVPA) as crucial for healthy ageing [5]. Indeed, regular physical activity is considered to be important for preventing sarcopenia development [6]. Importantly, the recently updated PA guidelines for older adults also recommend engagement in muscle strengthening activities (MSA), at least two times per week [5], supported by the well-documented myotrophic role of MSA [7,8,9] and its beneficial influence on fall risk and ability to perform activities of daily living [10,11,12].

Unfortunately, it is currently estimated that less than 20% of older adults adhere to the current PA guidelines including the MSA recommendation [13,14,15], with only about 13% of older adults reporting engagement in MSA at least twice a week [16].

Currently, there is a paucity of data on the potential health-related impacts of the MSA recommendation beyond those inferred by accumulating 150 weekly minutes of MVPA. Hence, the MSA recommendation has been referred to as the forgotten guideline [16]. A limited number of studies have previously reported on associated impacts of adherence to the MSA recommendation in addition to the MVPA recommendation in the guidelines on general and cardiovascular-related chronic conditions [14,17,18,19]. This highlights the beneficial effect of adhering to guidelines on both MSA and MVPA as compared with MVPA alone. However, any potential effects inferred by adding engagement in MSA on the top of adherence to the MVPA recommendation on sarcopenia risk in older adults remain to be elucidated.

Therefore, the aim of the present study was to determine whether or not engagement in MSA is linked to sarcopenia risk in older adults who meet the recommendation in the PA guidelines of 150 min of MVPA per week.

## 2. Materials and Methods

### 2.1. Participants

The present cross-sectional study included 193 community-dwelling older men (*n* = 71) and women (*n* = 122), 65–70 years old, recruited through local advertisement within the frame of the EURODIET project. Inclusion criteria included the following: absence of overt disease, cardiovascular, diabetes and psychiatric conditions; disability in respect to mobility; accumulating a weekly amount of at least 150 min of MVPA. All investigations were conducted in accordance with the principles set by the Declaration of Helsinki. All participants were provided written information about the study and written consent was obtained. The study protocol was approved by the regional ethics committee of Uppsala, Sweden.

### 2.2. Anthropometry

Body height was measured with a stadiometer (Seca, Hamburg, Germany) using standard procedures. Waist circumference (WC) was measured to the nearest 0.1 cm at the midpoint between the iliac crest and lower costal margin using the Seca 203 (Hamburg, Germany) circumference measuring tape. Body weight and skeletal muscle mass index (SMI) were assessed using bioelectrical impedance analysis (Tanita MC-780, Tanita Amsterdam, The Netherlands). Skeletal muscle mass (SMM) was calculated according to the equation of Janssen et al. [20], and then divided by body weight to obtain the skeletal muscle index (SMI, kg/BW).

### 2.3. Assessment of Adherence to PA Guidelines

Adherence to the PA guidelines regarding 150 weekly min of MVPA was assessed using the Actigraph GT3x (Actigraph, Pensacola, FL) accelerometer for a week, as previously described [21]. Accelerometer count cut-point for MVPA was >2019 counts per minute [22]. Participants accumulating an average of 22 min of MVPA per day (approximating 150 min per week) were classified as meeting the MVPA guideline. Adherence to MSA was assessed using the EPAQ2 questionnaire [23] where participants reported on duration and frequency of MSA during the last 12 months on the following activities: strength training, yoga types, rhythmic gymnastics/aerobic low and high intensity, water-based gym, own designed training, DVD-based exercises, motion gym, qigong, rubber band based, core workout, and sit ups. According to reported frequencies, participants were stratified based on whether they reported MSA at least twice per week (adherence) or not (no adherence).

### 2.4. Assessment of Adherence to Protein Intake

Adherence to recommended amounts of protein intake was assessed by a validated 84-item food frequency questionnaire (FFQ) [24]. Daily protein intake was normalized to bodyweight and expressed as g/bodyweight (BW). Cut-point for adherence to guideline on protein intake was set to 1.1 g/BW, in accordance to recommendations issued for older adults [25].

### 2.5. Assessment of Physical Function

Handgrip strength was assessed by standardized procedures using a Jamar handheld dynamometer (Patterson Medical, Warrenville, IL, USA). A five times sit-to-stand (5STS) was performed, whereby participants were instructed to start from a standing fully upright position and to sit down in a chair and repeat this sequence 5 times.

### 2.6. Sarcopenia Risk Score

A continuous clustered sarcopenia risk score (SRS) was created in accordance with the recent operational definition of sarcopenia by the European Working Group on Sarcopenia in Older People (EWGSOP) [1], including handgrip strength (HG), skeletal muscle index (SMI), and 5STS. The SRS is calculated by first standardizing (z-scores) muscle strength (handgrip), muscle mass (SMI), and physical performance (5STS) for men and women, separately. Thereafter, the three sex-specific standardized variables are summed, averaged, and merged into one final sex-adjusted SRS variable, as previously described [26,27].

### 2.7. Statistical Analyses

Data are presented as mean and standard deviation unless otherwise stated and checked for normality using visual inspection of probability plots, as well as conducting the Kolmogorov–Smirnov normality test. Differences between males and females were determined by either independent sample t-tests or Chi-square tests. Factorial analysis of variance (ANOVA) was employed to determine differences in sex-specific SRS between those reporting MSA at least twice a week (adherence) or not (no adherence). First, ANOVA models were adjusted by potential influence of age, use of prescribed medication (yes/no) and tobacco use (never, past use, current use), waist circumference (WC) based on established metabolic risk cut-points of ≥94 cm for men and ≥80 cm for women [28], and adherence (yes/no) to the recommendation on protein intake (1.1 g/BW). As there were no significant differences in SRS between groups of medication use or tobacco use, these two variables were omitted in final models to retain statistical power. The same analysis was performed on each separate component of SRS (SMI, HG, and 5STS). A priori power calculation showed that small to moderate effect sizes are detected with a power of ≥80% when based on our sample size and alpha level set to 0.05. All analyses were conducted using SPSS version 27.

## 3. Results

The general characteristics of the population are presented in Table 1. Men had significantly higher handgrip strength (*p* < 0.05) and SMI (*p* < 0.05) compared to women, with no differences observed in 5STS (Table 1). Additionally, a significant higher WC was found in men (Table 1). Twenty-four percent of men and 30% of women reported engagement in MSA at least twice a week with a weekly duration averaging 157 ± 75.6 and 174 ± 91.1 min for men and women, respectively. A total of 33% of the population adhered to the guideline for daily protein intake (1.1 g/BW) and 45% used prescribed medication. Finally, 5% and 50% were current or past tobacco users, respectively.

We investigated the related impact of adhering to MSA twice a week on SRS and its separate components (HG, SMI, and 5STS). The ANOVA revealed a significant main effect of MSA adherence on SRS (*p* < 0.05), whereas those reporting MSA twice a week had a significantly lower SRS compared to those who did not (Figure 1). Importantly, this finding was evident even after adjustment for adherence to the protein intake guideline and other covariates. Our data further showed a significant main effect of MSA adherence on SMI (*p* < 0.05) and 5STS (*p* < 0.05), with higher SMI and better 5STS performance in those adhering to the MSA guideline compared to those who did not (Figure 2a,b). Notably, no main effect of MSA adherence on handgrip strength was observed (0.001 ± 0.086 vs. 0.001 ± 0.132, *p* = 0.999).

In addition, given that older adults may still receive health benefits from lower volumes of MSA than currently recommended, we re-analyzed our data to investigate whether engagement in MSA at least once instead of at least twice a week impacts on SRS and its related components. Interestingly, a significant main effect of MSA on 5STS performance was observed (*p* < 0.05), where those reporting MSA at least once a week had a better functional performance (5STS) (−0.196 ± 0.104 vs. 0.209 ± 0.106, *p* = 0.005) as compared with those who did not. These main effects remained evident after adjustment for adherence to the recommended protein intake guideline and other covariates. In contrast to results when comparing groups who reported MSA at least twice a week, no differences in SRS (0.076 ± 0.069 vs. −0.107 ± 0.070, *p* = 0.053) and SMI (−0.002 ± 0.084 vs. 0.075 ± 0.086, *p* = 0.506) were revealed when comparing groups reporting MSA at least once a week or not. Finally, no difference in handgrip strength was observed between MSA groups (−0.031 ± 0.100 vs. 0.035 ± 0.102, *p* = 0.628). 

## 4. Discussion

The present study investigated whether or not engagement in MSA is linked to sarcopenia risk in physically active older adults. Here, we show, for the first time, that engagement in MSA according to current PA guidelines is related to a lower sarcopenia risk, with higher SMI and better physical performance in older adults who fulfil the PA guidelines on weekly MVPA. Notably, the beneficial impact of MSA was evident regardless of adherence to recommended amounts of protein intake.

Adherence to the MSA recommendation in the PA guidelines has previously been shown to infer reductions in cardiovascular and metabolic disease risks [14,18], as well as improvements in physical function [19]. Importantly, the present study specifically addresses the potential additional health benefits of MSA beyond adherence to the recommended amount of weekly MVPA. The combined impact on sarcopenia risk of adherence to PA guidelines, including both accumulation of weekly MVPA time and MSA at least twice a week, compared to adherence to the MVPA recommendation alone, is of great importance for the ageing population. Our findings show that engagement in MSA at least twice a week is related to a lower sarcopenia risk, where the beneficial impact was reflected by larger muscle mass and better physical performance. In contrast to typical endurance-type activities relying on oxidative capacity, MSA requires movements with high-force output, and when performed on a regular basis are known to stimulate muscle hypertrophy [7,8,9]. Moreover, regular MSA elicits neuromuscular adaptations related to motor unit recruitment [29,30], and thereby impacts on physical performance such as 5STS. The absence of an association between MSA adherence and handgrip strength may partly be explained by the fact that none of our study participants was sedentary, as they all adhered to the MVPA guideline, making it less likely that they would have poor overall muscle strength. Therefore, any potential beneficial impact of MSA on this muscle strength proxy is likely smaller than would have been expected in a sample of sedentary older adults at a higher risk of frailty. Additionally, the crude nature of handgrip as a marker of overall muscle strength in our population may further obscure detection of any MSA-related impact. Altogether, our study clearly highlights the clinical importance of weighing together several identified dimensions contributing to sarcopenia development in order to determine putative impacts of amounts and types of physical activity.

Notably, we further investigated whether engagement in MSA at least once a week, rather than at least twice a week, was related to lower sarcopenia risk. While beneficial effects were indicated on lower body performance, no concomitant impacts on sarcopenia risk, muscle mass, and handgrip strength were detected. The lack of an association with sarcopenia risk and muscle mass is likely explained by the reduced frequency cut-point (i.e., ≥1 instead of ≥2 times a week) directly affecting the weekly volume of MSA (weekly total, 70 min vs. 166 min). Indeed, a previous systematic review concluded that training twice a week promoted superior effects on measures of muscle hypertrophy as compared with once a week [7] and our findings highlight the importance for older adults to meet the MSA recommendations issued by major public health organizations. Taken together, our findings hold important public health implications. First, the fact that older adults who already adhere to the recommended amount of 150 min of weekly MVPA still benefit from engagement in MSA, puts special emphasis on this recommendation in the guidelines, sometimes labelled as the forgotten PA guidelines. Second, the findings emphazise the independent role of engagement in MSA alongside both general MVPA time and dietary protein intake for prevention of sarcopenia in older adults.

Our study findings are strengthened by the use of validated tools for assessment of adherence to PA guidelines, including accelerometery-determined MVPA time and the well-established EPAQ2 questionnaire. Another strength is the inclusion of important variables known to have an influence on measures of sarcopenia risk when determining the independent role of MSA. By adjusting for adherence to the protein intake guideline, we accounted for a powerful driver of muscle hypertrophy, as it has recently been linked to both muscle mass [31] and sarcopenia risk [27]. Moreover, we further considered the potential impact of central obesity on progression of sarcopenia [32,33]. The present study also took advantage of the latest operational definition of sarcopenia, which emphasizes the integration of muscle strength and mass with functional performance as an indicator of severity [1]. However, the study is not without limitations. Due to its cross-sectional nature, caution should be taken when interpreting the direction of the relationship. Furthermore, the study sample was comprised of older adults who adhered to the PA guideline on weekly MVPA time in order to address the additional role of MSA; however, given this specific sample, it is unlikely to be representative for broader groups of older adults with lower PA levels or diagnosed diseases. It should be noted that classification of MSA may vary among studies, as different activities involve various degrees of muscle force generation. However, the activities classified as MSA are similar to those defined in previous studies [14,16,19].

## 5. Conclusions

The present study shows that engagement in MSA at least twice a week is linked to sarcopenia risk, with larger muscle mass and better physical performance in older adults who already accumulate 150 weekly minutes of MVPA. This beneficial impact is independent of adherence to the recommended protein intake, which further reinforces the promotion of the recommendation of MSA at least twice a week, along with the general accumulation of weekly MVPA time to combat sarcopenia progression in older adults.

## Figures and Tables

**Figure 1 ijerph-18-00989-f001:**
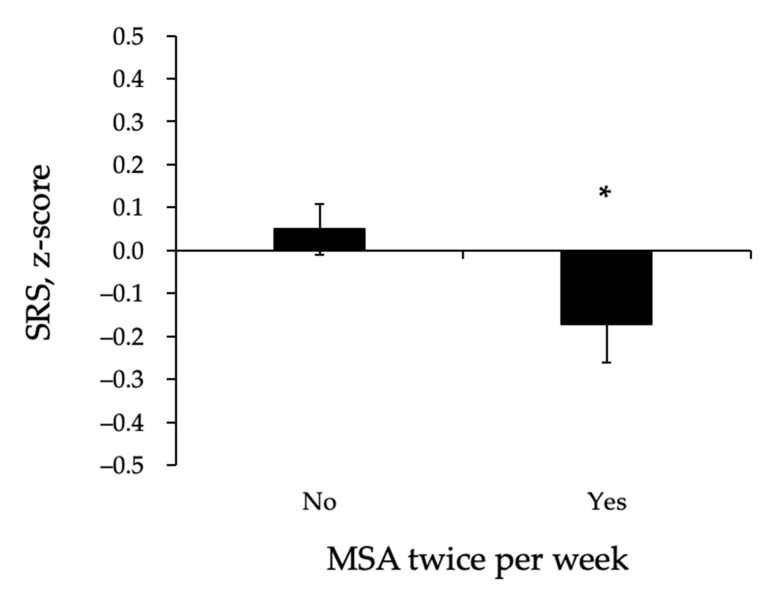
Sarcopenia risk score in those reporting MSA twice a week (yes) and those who did not (no). MSA, muscle strengthening activities. Data are means ± SEM adjusted for age, gender, waist circumference, and protein intake. * *p* < 0.05.

**Figure 2 ijerph-18-00989-f002:**
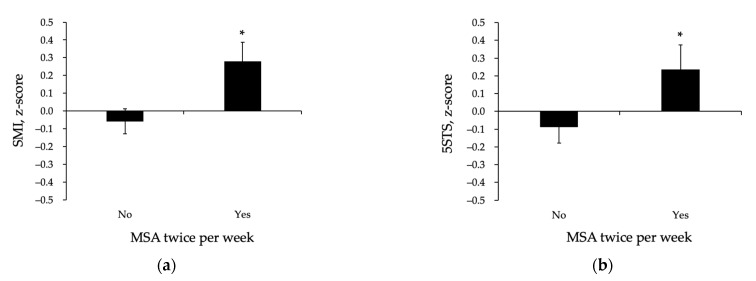
Skeletal muscle mass index (**a**) and five times sit-to-stand (5STS) (**b**) in those reporting MSA twice a week (yes) and those who did not (no). MSA, Muscle strengthening activities; 5STS, five times sit to stand. Data are means ± SEM adjusted for age, gender, waist circumference, and protein intake. * *p* < 0.05.

**Table 1 ijerph-18-00989-t001:** General characteristics of the study population.

	Male	Female
*n*	71	122
Age, y	67 ± 2	67 ± 2
**Body Composition**		
Height, cm	178.1 ± 6.0	165.0 ± 5.3 *
Weight, kg	80.0 ± 1.0	63.7 ± 9.2 *
WC, cm	93.7 ± 10.0	79.1 ± 8.5 *
**Sarcopenia risk components**		
SMI, % BW	34.5 ± 3.2	26.8 ± 3.4 *
Handgrip, kg	44.1 ± 7.0	28.2 ± 5.2 *
5STS	10.0 ± 2.0	10.2 ± 2.3

Data are presented as mean ± SD or are otherwise indicated. WC, waist circumference; BW, body weight; kg, kilogram; SMI, skeletal muscle index; 5STS, 5-time sit-to-stand; HG, handgrip. * *p* < 0.05 vs. male.

## Data Availability

Data supporting reported results are available upon reasonable request due to ethical principles.

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
