# Peer review of "Engagement in Muscle-Strengthening Activities Lowers Sarcopenia Risk in Older Adults Already Adhering to the Aerobic Physical Activity Guidelines"

_ijerph, 2021, doi:10.3390/ijerph18030989_

Round 1
Reviewer 1 Report
The submitted article presents a topic with some relevance, however the way it was worked does not seem to be the best. It needs some improvements in the description of the methods and major changes in the presentation of the results. The points that need to be changed are described below:
Materials and methods:
Lines 60 to 68 - In point 2.1, the type of epidemiological study must be identified.
Line 63 - Why only participants aged 65 to 70 years were selected? Authors must justify this fact. What is the criterion used to exclude participants over 70 years old?
Line 70 - the equipment (brand, model and place of manufacture) used to measure weight and height is not indicated. Needs to be indicated.
Line 99 - SRS was created based on which Guidelines? A reference is missing.
Line 113 - What program was used to calculate the minimum sample size? And what was the value obtained? Results:
Line 122 - Table 1 should be reformulated. It is not clear which test was applied to make a comparison between genders. Was the variables distribution normality verified? What test was used as a result of this conclusion? The p-value results should be presented in the table. All acronyms used in the table must be described in the footer. The Table should be organized between two groups (those who reported joining the MSA and those who did not report joining) and not just between genders. Results on body composition should also be presented in this or in another table, as they are not described in the results.
The results of the 84 items validated FFQ are also not presented although their application has been described in lines 89 and 90 of the methods. The results obtained for the daily protein intake and its adherence to the guidelines (1.1 g / BW) must be presented.
The figures presented do not show ANOVA results. Lines 126 to 128 describe the results of ANOVA using Figure 1 as a reference. The same is done for lines 129 to 133 for figure 2 (a) and (b). The figures show only Z-score results and not the results of applying the ANOVA test. I recommend to make a Table with all the ANOVA results and respective p-values. I also recommend to performed the ANCOVA test to obtain the adjusted means for the confounding variables described in figure 2, as well as, a logistic regression that allows to estimate the probability of risk of sarcopenia for Older adults who are already adhering to the aerobic physical activity guidelines in comparison to those with no adherence.
The Discussion should be reformulated according to the results obtained after applying the suggested tests.
The Summary also needs improvement. The type of study should be identified and the results must be described in a more objective way (as indicated above).
Author Response
We thank the reviewer for constructive remarks that substantially improved the quality of the manuscript. We have addressed all comments and made appropriate changes. Please find below a point-by-point response to all comments.
Materials and methods:
Lines 60 to 68 - In point 2.1, the type of epidemiological study must be identified.
The cross-sectional study design is now stated (page 2, lines 61).
Line 63 - Why only participants aged 65 to 70 years were selected? Authors must justify this fact. What is the criterion used to exclude participants over 70 years old?
The aim of this study was to determine the additional impact of MSA adherence on the top of adhering to the aerobic part of the PA guideline. This approach requires a selection of the older population that has a physically active lifestyle. For this reason, the narrow age range (65-70 years) is more likely to ensure a study population of healthy older men and women without presence of manifest disease states, which in turn reflects a growing part of the global population at these particular ages. Moreover, gathering data on populations within this age range with no manifest diseases provides possibility to better understand trajectories of age-related loss of physical function.
Line 70 - the equipment (brand, model and place of manufacture) used to measure weight and height is not indicated. Needs to be indicated.
Information about equipment used is now provided (page 2, line 70-74).
Line 99 - SRS was created based on which Guidelines? A reference is missing.
References have now been added (Page3 , line 102).
Line 113 - What program was used to calculate the minimum sample size? And what was the value obtained?
Power calculation was conducted using the software G*power adapted for F-test family (e.g. factorial ANOVA with fixed effects). Effects sizes around 0.2 were obtained based on our sample size, with a power of 80% and alpha set to 0.05.
Results:
Line 122 - Table 1 should be reformulated. It is not clear which test was applied to make a comparison between genders. Was the variables distribution normality verified? What test was used as a result of this conclusion? The p-value results should be presented in the table. All acronyms used in the table must be described in the footer. The Table should be organized between two groups (those who reported joining the MSA and those who did not report joining) and not just between genders. Results on body composition should also be presented in this or in another table, as they are not described in the results.
Independent samples t-test (continuous outcomes) and chi-square test (proportions) were used for sex-comparisons (page 3, line 108-109). Normality was checked using visual inspection of probability plots (q-q-plot) as well as conducting a normality test (Kolmogorov-Smirnov). This has now been clarified in the manuscript (page 3, line 107-108). In line with common standard procedures, significant differences between sexes are indicated by an asterisk, denoting p-values less than 0.05. All acronyms used in the table are now described in the footer (page 4, line: 129-131).Table 1 presents general characteristics of the study population (e.g. height, weight), which are heavily influenced by biological sex, and therefore stratified by sex. However, regarding the primary outcome of the study, which is a continuous standardized sarcopenia risk score (SRS) already normalized for sex differences, we find it more appropriate to present results on the impact of MSA adherence on SRS based on the whole population in figure 1. By doing so, table 1 will describe general characteristics in a well-defined population of older men and women. Results on body composition (height, weight, waist circumference, SMI) are presented in table 1.
The results of the 84 items validated FFQ are also not presented although their application has been described in lines 89 and 90 of the methods. The results obtained for the daily protein intake and its adherence to the guidelines (1.1 g / BW) must be presented.
We simply used the FFQ to assess adherence to protein guidelines. We have now added information about adherence to the protein guideline (page 3, line 126)
The figures presented do not show ANOVA results. Lines 126 to 128 describe the results of ANOVA using Figure 1 as a reference. The same is done for lines 129 to 133 for figure 2 (a) and (b). The figures show only Z-score results and not the results of applying the ANOVA test. I recommend to make a Table with all the ANOVA results and respective p-values. I also recommend to performed the ANCOVA test to obtain the adjusted means for the confounding variables described in figure 2, as well as, a logistic regression that allows to estimate the probability of risk of sarcopenia for Older adults who are already adhering to the aerobic physical activity guidelines in comparison to those with no adherence.
Thank you for this question. In figure 1 and 2 we report adjusted means that results from the factorial ANOVA models and which are calculated after covariates have been taken into account. We also denote significant differences from main effects by asterisks in the figures. Hence, data presented in these figures results from the factorial ANOVA models after adjustment for covariates.
Given that our selected sample of older men and women does not fulfill established criteria for manifest sarcopenia, stratification of participants into groups with and without sarcopenia cannot be made. Subsequently, analysis based on logistic regression models requiring a categorical outcome variable would not be possible. Hence, the results and following conclusions presented in the manuscript remain unaltered.
The Summary also needs improvement. The type of study should be identified and the results must be described in a more objective way (as indicated above).
Assuming that ‘The Summary’ refers to the Abstract, we have now added the requested information regarding study design (page 1, line 14). As for the reviewer request to describe the results in a more objective way, we state that ‘Guideline adherence to MSA twice a week was related to a significantly (p < 0.05) lower SRS compared to those who did not. This finding was evident after adjustment for adherence to protein intake guideline and abdominal obesity’ (page, 1 lines 21-23), which we hope will be viewed as an objective summarization of the study outcomes.

Reviewer 2 Report
Very interesting topic but some major comments:
- It is unclear why the authors used a sarcopenia risk score they developed. Why not perform these analyses on the status of sarcopenia (dx yes or dx no)? In addition, little information is provided on how the authors developed the score. This is a point that needs to be explored further.
- Stats in more detail: how is normality verified? No non-parametric test? why and how these adjustment variables were chosen. Why not perform correlation or association measures (multiple regression)? This greatly reduces the impact that the results could have.
- Dose-response relationship: the more MSA is done the higher the sarcopenia risk score?
- The inclusion criteria are strict (i.e., no disease, which is uncommon beyond the age of 65) and therefore does not represent the general older population living in the community. In addition, sampling is on a voluntary basis. How does this impact the results?
- What would the authors' findings be in other settings (i.e., nursing home)?
- Limit design of the study (which should ideally be in the title)
- Important confounding factors, most likely impacting sarcopenia, were not taken into account: overall nutritional status, number of drugs, number of co-morbidities, smoking status, physical activity level, cognitive status, etc., which largely limits the scope of the results.
Author Response
We thank the reviewer for constructive remarks that substantially improved the quality of the manuscript. We have addressed all comments and made appropriate changes. Please find below a point-by-point response to all comments.
It is unclear why the authors used a sarcopenia risk score they developed. Why not perform these analyses on the status of sarcopenia (dx yes or dx no)? In addition, little information is provided on how the authors developed the score. This is a point that needs to be explored further.
Thank you for this question. Our selected sample of older men and women does not fulfill established criteria for manifest sarcopenia. Therefore, the analyses are based on a continuous score encompassing the three dimensions (muscle strength and quantity, and functional performance) that are identified in the recent operational definition of sarcopenia by the European Working Group on Sarcopenia in Older People (EWGSOP). (Page 3, line 100-104)
Stats in more detail: how is normality verified? No non-parametric test? why and how these adjustment variables were chosen. Why not perform correlation or association measures (multiple regression)? This greatly reduces the impact that the results could have.
Normality was checked using visual inspection of probability plots (q-q plot) as well as conducting a normality test (Kolmogorov-Smirnov). This has now been clarified in the manuscript (page 3, line: 106-108). As noted in the manuscript, chi-square tests (non-parametric test) were conducted for comparison of proportions of MSA adherence between the sexes. Given the specific aim of this study (determine differences in sarcopenia risk between MSA adherence groups (Yes/No)), measures of associations based on continuous data variables, such as correlation coefficients, become less appropriate. Instead, a factorial ANOVA was chosen as such analysis determines the effect on a continuous dependent outcome (e.g. SRS) by fixed factors (e.g. MSA Yes/No).
Dose-response relationship: the more MSA is done the higher the sarcopenia risk score?
As stated previously, the scope of the analysis was to determine the potential effects of adhering to stated PA guidelines about engagement in MSA (i.e. twice per week; Yes/No) and as such, explorations of potential dose-response relationships go beyond the scope of this study. Notably, we present data based on adherence to MSA once per week instead of twice per week and show that beneficial impacts of MSA adherence on SRS and its components were attenuated.
The inclusion criteria are strict (i.e., no disease, which is uncommon beyond the age of 65) and therefore does not represent the general older population living in the community. In addition, sampling is on a voluntary basis. How does this impact the results?
The aim of this study was to determine the additional impact of MSA adherence on the top of adhering to the aerobic part of the PA guideline. Thus, a selection of the older population that has a physically active lifestyle was required. Such a sample may not be representative of broader populations of older men and women, which is also acknowledged in the manuscript (page 6, line 214-215). Nevertheless, our study sample reflects a growing part of the global population at these ages and gathering data on such populations provides possibilities to better understand trajectories of age-related loss of physical function and the preventive effects of physical activity on sarcopenia progression.
In addition, the fact that sampling was conducted on a voluntary basis follows ethical standards set by the declaration of Helsinki, endorsed by the global research community.
What would the authors' findings be in other settings (i.e., nursing home)?
Given that populations in nursing homes are likely to be frail and diagnosed with multi-comorbidities, it is highly unlikely that they would adhere to the weekly 150-minute guideline of aerobic moderate-to-vigorous physical activity as our study population did. Therefore, studies conducted on targeted populations with poor health status (i.e. nursing homes) would need a different approach.
Limit design of the study (which should ideally be in the title)
A section devoted to limitations of the study, including design, is presented in page 6 line 211-218
Important confounding factors, most likely impacting sarcopenia, were not taken into account: overall nutritional status, number of drugs, number of co-morbidities, smoking status, physical activity level, cognitive status, etc., which largely limits the scope of the results
Thank you for this question. Indeed, adjustment for known confounders is an important step in studies such as ours. Therefore, all analyses were adjusted by basic characteristics (age, sex), lifestyle factors (physical activity, diet (protein intake)) as well as biological risk factors (abdominal obesity by waist circumference). Notably, we used protein intake as a covariate, rather than any composite measure of overall nutrition status when modelling the impact of MSA adherence on SRS. This since an adequate protein intake is regarded of particular importance for the regulation of muscle quantity and quality during aging (Montiel-Rojas et al. Nutrients 2020, 12(12), 3601). Moreover, all participants were free of overt disease to avoid potential confounding from number of co-morbidities. In accordance with the reviewer’s suggestion, we have now extended our analysis by checking for potential confounding by medication use and tobacco use. As these two variables did not influence on SRS, they were omitted from the final model (page 3, lines 114 -116).

Round 2
Reviewer 1 Report
The arThe article improved significantly. The suggested changes were made and some aspects were clarified that were not quite clear. Thus, I believe that this article already meets the necessary conditions to be published.
Author Response
We thank the reviewer for endorsing publication of our manuscript.
Reviewer 2 Report
The authors have significantly improved the manuscript, which is suitable for publication after a final minor revision:
- It is essential that the authors be more transparent with the score used to make the research reproducible. How is it calculated? Is it valid? Has it been tested against a gold standard? What sensitivity? What specificity? What is the weight of each variable? What scale (Z score), but how to interpret it?
- The authors do not want to determine a dose-response relationship, we must at least talk about this limit in the discussion.
Author Response
- It is essential that the authors be more transparent with the score used to make the research reproducible. How is it calculated? Is it valid? Has it been tested against a gold standard? What sensitivity? What specificity? What is the weight of each variable? What scale (Z score), but how to interpret it?
Reply: Thank you for this question. While no gold standard method to assess sarcopenia risk currently exists, single components of sarcopenia have been frequently used in previous research to denote sarcopenia risk (e.g., muscle mass or muscle strength). However, in the recent operational definition of sarcopenia by the European Working Group on Sarcopenia in Older People (EWGSOP), three components – muscle strength, muscle mass and physical performance - are identified as key screening determinants that should be jointly evaluated when determining sarcopenia. Therefore, encompassing these three physical determinants of sarcopenia into one composite score will imply good construct validity in term of sarcopenia risk and likely better reflect individual variations in sarcopenia progression compared to using any single dimension alone.
The sarcopenia risk score (SRS) is calculated by first standardizing (z-scores) muscle strength (handgrip), muscle mass (SMI) and physical performance (5STS). This is done separately for men and women. Thereafter, the three standardized variables are equally summed and averaged into one composite score (SRS), and finally merged into one sex-adjusted SRS variable, where higher values denotes higher risk of sarcopenia.
Notably, we have used the same procedure for calculating SRS in recent studies linking aspects of dietary intakes to sarcopenia risk (Montiel-Rojas, et al. Nutrients. 2020 Nov 24;12(12):3601; Montiel-Rojas, et al. Nutrients. 2020 Oct 9;12(10):3079).
For interpretation purposes, a higher SRS denotes a higher sarcopenia risk compared to individuals with lower SRS, as a higher SRS implies that levels of the three key determinants are generally lower compared to individuals with lower SRS. Differences in SRS between groups are standardized effects sizes, as expressed in standard deviations from the mean (with mean value set as 0).
Regarding sensitivity and specificity asked by the reviewer, please consider that the SRS is a continuous variable denoting variation in sarcopenia risk within a given sample, and not a tool developed to classify participants into sarcopenia (yes/no) by any fixed cut points. Please also consider that none of our participants fulfill established criteria for manifest sarcopenia.
In accordance with reviewer comment, calculation of SRS has now been further described in the manuscript (page 3, line: 110-115)
- The authors do not want to determine a dose-response relationship, we must at least talk about this limit in the discussion.
Reply; We agree with the reviewer that increased knowledge about dose-response relationships between PA and different health outcomes are an important scope of research. However, we kindly like to note that the aim of the present manuscript addresses whether or not adherence to guidelines on muscle-strengthening activities, assessed as a fixed frequency of at least twice per week (yes/no), provides beneficial effects on top of fulfilling the aerobic-exercise guideline of 150 weekly minutes of moderate-to-vigorous PA. Thus, given our aim questions regarding dose-response relationships, preferably requiring continuous PA variables, cannot be addressed. Importantly, this should not be viewed as a study limitation as determination of dose-response relationships never was part of the study aim.